# Interaction Effects of Physical and Psychosocial Working Conditions on Risk for Sickness Absence: A Prospective Study of Nurses and Care Assistants in Sweden

**DOI:** 10.3390/ijerph17207427

**Published:** 2020-10-12

**Authors:** Magnus Helgesson, Staffan Marklund, Klas Gustafsson, Gunnar Aronsson, Constanze Leineweber

**Affiliations:** 1Division of Insurance Medicine, Department of Clinical Neuroscience, Karolinska Institutet, SE-171 77 Stockholm, Sweden; magnus.helgesson@ki.se (M.H.); staffan.marklund@ki.se (S.M.); klas.gustafsson@ki.se (K.G.); 2Department of Psychology, Stockholm University, SE-106 91 Stockholm, Sweden; gunnar.aronsson@psychology.su.se; 3Department of Psychology, Stress Research Institute, Stockholm University, SE-106 91 Stockholm, Sweden

**Keywords:** nurses, care assistants, physical work, psychosocial work, synergy index, sickness absence

## Abstract

Employees in health and social care are often simultaneously exposed to both physical and psychosocial challenges that may increase their risk for sickness absence. The study examines interaction effects of physical and psychosocial work conditions on the future risk for sickness absence among nurses and care assistants in Sweden. The study was based on 14,372 participants in any of the Swedish Work Environment Surveys conducted during the years 1993–2013 with linked register information on background factors and compensated sickness absence. Adjusted hazard ratio (HR), stratified by occupation, and measures of additive interaction effects were estimated. The combinations of high psychosocial job demands and heavy physical work and strenuous postures, respectively, significantly increased the risks for sickness absence among nurses (HR 1.43; CI 1.09–1.88 and HR 1.42; CI 1.16–1.74, respectively), as well as among care assistants (HR 1.51; CI 1.36–1.67 and HR 1.49; CI 1.36–1.63, respectively). The combinations of low job control and both heavy physical work (HR 1.44; CI 1.30–1.60) and strenuous postures (HR 1.42; CI 1.30–1.56) were also associated with excess risk for sickness absence among care assistants. We also found interaction effects among care assistants but not among nurses. The results indicate that the high sickness absence rate among care workers in Sweden can be reduced if the simultaneous exposures of high psychosocial and high physical challenges are avoided. Management policies for reduced time pressure, improved lifting aids, and measures to avoid awkward work postures are recommended. For care assistants, increased influence over work arrangements is likely to lower their sickness absence risk.

## 1. Introduction

During the last 25 years, health and care workers in Sweden have experienced increasing psychosocial challenges at work but without any corresponding reduction in the exposure to physical factors. This may be one of the main reasons behind their high rates of sickness absence [1]. In order to fully understand why the rate of sickness absence is high among health care employees, it is important to study the combined effect, as both physical and psychosocial work exposures are highly prevalent within the health care sector. Particularly interesting is the degree to which the effects of two different exposures are higher than the added effect of each of them, i.e., an additive interaction effect.

An early study of potential interaction effects between physical and psychosocial work factors was a Danish investigation of accidents among slaughterhouse workers that found that the accident rate was associated with high job strain in combination with heavy physical demands [2]. Additionally, a later follow-up showed the same negative effect of job strain in combination with hard physical work on sickness absence in the same occupational group [3]. These findings imply an interaction between heavy physical work, job strain, and health risks among employees who are subject to such simultaneous exposures.

The combination of high psychosocial demands and low control at work was developed by Karasek and Theorell [4,5], and its effect on health and sickness absence has been investigated in several studies [6,7,8]. In accordance with Karasek’s and Theorell’s model [5] and Rothman’s synergy index [9], the assumption of the present study is that combinations of unfavorable physical and unfavorable psychosocial working conditions reinforce each other and cause an increasing risk of sickness absence that is larger than the sum of the risks of each of them. For example, time pressure in urgent situations may cause that ergonomically favorable techniques and practices are not used.

A number of studies of different occupational groups reported additive interaction effects of physical and psychosocial job factors on low back symptoms and absenteeism [10,11,12,13]. Some studies of different occupations have also found that the combination of different aspects of physical and psychosocial load affected health or sickness absence, but in many cases, these interaction effects were different among women and men [14,15,16,17,18,19,20,21]. However, a few studies of mixed occupations did not find any interaction effects of physical and psychosocial exposures for musculoskeletal disorders or sickness absence [22,23].

Research reviews and individual studies among health and social care workers have shown that heavy lifting, twisting, and bending often are combined with urgent time pressure and low possibilities to determine the work pace, conditions known to increase the risk of musculoskeletal disorders, sickness absence, and disability [24,25,26,27,28,29,30,31,32,33]. However, the effect of the combined exposure of psychosocial and physical risk factors on future health and sickness absence among health and care employees is rarely studied in terms of interaction effects; instead, the focus has been on single factors [34,35,36].

The results of studies conducted among health and social care workers of interaction effects regarding physical and psychosocial risk factors show partly different results. A study of employees in home care reported an interaction effect for shoulder and neck pain for the combination of strenuous work postures and low job control [24], and an Iranian study among nursing staff showed that the combination of shift work and straining physical work increased the risk for low back pain more than the sum of each factor [25]. A study of hospital employees demonstrated that simultaneous exposure to emotional demands, time pressure, and cognitive demands had an additive disadvantageous effect on sleep, exhaustion, and job satisfaction [26], and a study by our research team found additive interaction effects between combined physical and psychosocial exposures and future disability pension among Swedish health and care workers [28]. However, no interaction effect was found between psychological and mechanical workloads and musculoskeletal symptoms in a study of female childcare workers in Sweden [37]. Despite the fact that health and care workers often report simultaneous exposure to physical and psychosocial risks, no study has been found on the interaction effects of physical and psychosocial exposures on sickness absence.

The present study had two aims. The first aim was to study the combinations of simultaneous exposure to different indicators of physical and psychosocial job factors for the risk of future sickness absence among nurses and care assistants in Sweden. The second aim was to estimate to what degree such combinations included additive interactions in the sense that the effect was larger than the sum of each of the components in the combination.

## 2. Materials and Methods

### 2.1. Data Materials and Participants

The main source of information was the Swedish Work Environment Survey (SWES), which has been conducted every second year since 1989 [38]. The SWES covers a broad range of work conditions and is based on representative samples of the Swedish employed population aged 16 to 64. The survey starts with a telephone interview, which is followed up by a postal survey. The survey questions are supposed to give an objective description of the work environment conditions, and validity and reliability of the variables in SWES were used [38]. Different methods were used to test the validity and reliability of the used variables in the SWES. One was to test the questions at the workplace, where the actual conditions were known and could be compared to other types of information, such as administrative and technical information. Additionally, a number of validation studies were conducted, where responses to different formulations of questions were used and compared [39].

The response rates of the SWES varied between 66% and 89 %. In this study, data from 11 iterations of the survey between 1993 and 2013 were used and complemented with information on sociodemographic factors and subsequent compensated sickness absences 1994–2016 from the Longitudinal Integrated Database for Health Insurance and Labor Market Studies (LISA), hosted by the Statistics Sweden register (www.scb.se).

Based on the Swedish Standard Classification of Occupations (SSYK, www.scb.se), participants from two occupational groups were selected. The first occupational category was called “Nurses”. It consisted of specialized and nonspecialized nurses (SSYK 223 and 323; *n* = 2716). The second occupational group was named “Care assistants”. It included assistant nurses, hospital ward assistants, and home-based personal care workers and assistants in child care (SSYK 513; *n* = 11,650). Ethical approval for the study was obtained by the Regional Research Ethics Board in Stockholm (2018/223-31/5).

### 2.2. Measurements

#### 2.2.1. Outcome Variable

The outcome variable was the total number of net days of compensated sickness absence during three calendar years following the year of participation in the SWES survey. The reason for the three-year period was a balance between two principles. One was to get enough numbers of sickness absent individuals. The other was to avoid a too long period as working conditions change over time. Only information on compensated sickness absence covered by the Swedish Social Insurance Agency was available in the population register. That is, no information on sickness absences during the first two weeks of a spell was available. Consequently, in our study, all individuals with at least one day of compensated sickness absence were absent in total for at least 15 days, while individuals with no registered sickness absence could have had up to 14 days of sickness absence.

#### 2.2.2. Physical and Psychosocial Exposure Variables

Since they first started in 1989, the SWES surveys have had two blocks of questions concerning exposure to physical and psychosocial risks at work, respectively. The questions and response alternatives largely remained the same. The selected items in this study concerned heavy physical work, strenuous job postures, psychosocial job demands, job control, and support from supervisors or colleagues. The items and the response alternatives were described in detail in a previous study from the research team [40]. In the following presentation, the official English translation of the wording of the questions was used (available at Statistics Sweden, http://www.scb.se) and the cut-offs between exposed and not exposed that were used in this study are presented in parenthesis.

The two aspects of physical exposure used in the study were heavy physical work and strenuous work postures. Both were based on three items, and the responses were used to form dichotomized indicators.

As indicators of heavy physical work, three different items were chosen. To indicate the most adverse conditions, the response alternatives were dichotomized closest to the upper quartile.

-Are you required to lift at least 15 kg at a time several times per day? (Exposed: Yes (≥1 out of every 5 days) and Not exposed: No (<1 out of every 5 days).-Does your job mean that your work is purely physical, i.e., do you put in more physical effort than you do when you walk, stand, and move in the usual way? Exposed: Yes (≥1/2 of the working time) and Not exposed: No (≤1/4 of the working time).-Do you exert yourself so much that you breathe faster? Exposed: Yes (≥1/4 of the working time) and Not exposed: No (≤1/10 of the working time).

Respondents who reported adverse conditions to at least two of the three items were classified as exposed to heavy physical work in a joint measure (Cronbach’s alpha = 0.84).

As indicators of strenuous work postures, three different items were chosen. To indicate the most adverse conditions, the response alternatives were dichotomized closest to the upper quartile.

-Do you bend or twist yourself in your work in the same way repeatedly in an hour, for several hours during the same day? Exposed: Yes (every day) and Not exposed: No (≤1 out of every 2 days).-Do you work bend forward, without supporting yourself with your hands or arms? Exposed: Yes (≥ 1/4 of the working time) and Not exposed: No (≤1/10 of the working time).-Do you work in a twisted position? Exposed: Yes (≥1/4 of the working time) and Not exposed: No (≤ 1/10 of the working time).

Respondents who reported adverse conditions to at least two of the three items were classified as exposed to strenuous work postures (Chronbach’s alpha = 0.80).

For psychosocial job demands, three items were chosen. To indicate the most adverse conditions, the response alternatives were dichotomized closest to the upper quartile.

-“Is your work so stressful that you do not have time to talk or even think about something other than work?” (Exposed: ≥3/4 of the time and Not exposed: ≤1/2 of the time.)-“Does the work require your full attention and concentration?” (Exposed: ≥3/4 of the time and Not exposed: ≤1/2 of the time.)-“Do you have so much work that you must miss lunch, work late, or take work home?” (Exposed: ≥1 day of 2 and Not exposed: ≤1 day per week)

Respondents who reported adverse conditions to at least two of the three items were classified, as exposed to high psychosocial demands in a joint measure. (Cronbach’s alpha = 0.57).

As an indicator of psychosocial job control, three items were used. To indicate the most adverse conditions, the response alternatives were dichotomized closest to the upper quartile.

-“Do you have the opportunity to determine your work pace?” (Exposed: ≤1/10 of the time and Not exposed: >1/10 of time.)-“Are you able to determine when various working duties are to be carried out (for example, by choosing to work a bit faster on some days and taking it easier on other days)?” (Exposed: No, not at all and Not exposed: Always, mostly)-“Do you participate in decisions on the arrangement of your work (e.g., what is to be done, how to do it, or who will work with you)?” (Exposed: No, not at all and Not exposed: Always, mostly, and mostly not)

Based on the responses to the three questions, a factor with low and high exposure was created, where high exposure means that two or three items indicated low job control (Cronbach’s alpha = 0.72).

In order to test for the potential combination effects and additive interaction effects, the dichotomized categories (1) heavy work, (2) strenuous postures, (3) high job demands, and (4) low job control were paired into four combinations: (1) high demands and heavy work, (2) high demands and strenuous postures, (3) low control and heavy work, and (4) low control and strenuous postures.

#### 2.2.3. Potential Confounders

Sex and age at interview (16–29, 30–39, 40–49, and 50–64 years); education (≤9 years, 10–12 years, and >12 years of education); country of birth (born in Sweden vs. foreign-born); and sector of employment (public sector (national, regional, or local authorities) and private sector) served as adjusting variables. All these variables concerned the year of the individuals’ participation in the SWES survey and were obtained from the LISA database.

### 2.3. Statistical Analyses

The individuals who participated in any of the Swedish Work Environment Survey (SWES) from 1993 to 2013 were successively added to the study cohort. The follow-up of sickness absence for each sub-cohort started on the first of January in the year after the interview and included the years 1994–2016. All analyses were stratified on two occupation groups.

The statistical examinations were conducted in two steps. In the first step, the associations of the combinations of physical and psychosocial working conditions and sickness absence were studied, adjusting for age at interview (one-year intervals) and year of interview (Model 1). Additionally, the associations were analyzed by an extension of the adjustment to also include sex, education, country of birth, and sector of employment (Model 2). Cox’s proportional hazards regression analyses were used to calculate the hazard ratios (HRs), and 95% confidence intervals (CI) were estimated. Separate analyses were conducted for the two occupational groups. Furthermore, a measure of internal consistency between items, the Cronbach’s alpha, was also calculated.

In the second step, possible additive interaction effects (synergetic) of physical work items and psychosocial working conditions on the risk for sickness absence were tested. These two dimensions may interact in different ways, but, specifically, we expected that the exposure to both negative physical and psychosocial conditions may increase the risk of sickness absence more than the sum of the two [41,42,43].

In order to estimate the existence of such additive interactions, the HRs of the two factors acting alone were compared with the HRs for joint exposure. Three different measures of interactions were calculated [41,44,45]. The main test statistic was the synergy index (S) with the formula S = (HR11 − 1)/((HR10 − 1) + (HR 01 − 1)).

Further, relative excess risk due to interaction (RERI = HR11 − HR10 − HR01 + 1) [46,47] and the attributable proportions (AP = RERI/HR11) were computed [41,44,45], where HR is the hazard ratio, and the 0/1 indicates the absence/presence of the risk factor. The delta method was used for the calculation of CIs of these three interaction measures [44,48].

Thus, the synergy index (S) is the ratio of the observed effect with the combined exposure to two risk factors in relation to the effect of independent exposures that are not reinforcing each other. A synergy index (S) and its CI above 1 indicate that the effect of two factors is larger than the sum of each of them, while a synergy index below 1 indicates that the two factors are antagonistic and neutralize each other [9]. RERI is a measure of the share of the total excess effect that is due to interaction. The attributable proportion is a measure of the share of sickness absence in the study group that could be prevented if the effect of the interaction of the two factors could be prevented. RERI and AP are both >0 if there is an interaction. All statistical analyses were conducted with SAS, version 9.4., statistical software (SAS Institute, Inc., Cary, NC, USA) using the PHREG procedure.

## 3. Results

As shown in Table 1, a higher proportion of female health and care employees in Sweden (36.5%) than of male (22.6%) had sickness absence during the follow-up period of three years. Both among women and men, a larger proportion of older individuals had been sickness-absent. The differences between the two occupational groups regarding sickness absence were smaller among men than among women (Table 1).

Information on background factors and working conditions for those with and without sickness absence during the follow-up period for the two occupational groups is presented in Table 2. The female share of nurses was 91.1% among those without sickness absence and 94.4% among those with sickness absence (Table 2). For care assistants the corresponding figures were 89.5% and 94.6%, respectively. Among nurses, there were no educational differences between those with and those without sickness absence, while cares assistants with longer educations had lower risks of sickness absence (18.0% and 14.1%, respectively) (Table 2).

The share of nurses exposed to do heavy work was slightly higher among those with sickness absence compared to those without absence (14.7% and 13.5%, respectively) (Table 2). The exposure to strenuous postures was more common among nurses with absence compared to those without absence (28.9% and 20.0%, respectively). For care assistants, there were differences in exposure to heavy work and strenuous postures between the sickness absent and those not absent, but the share exposed was generally higher (36.5% versus 28.3% and 45.0% versus 33.9%), Table 2. High job demands and low job control were more common among the sickness absent in both occupational groups than among those not absent (Table 2).

The differences in the background factors between those with and those without sickness absence are relatively small, while the differences with regard to work environment exposures between sickness absent and not sickness absent are larger. The differences indicate that sickness absence levels are affected by these factors and that they must be controlled for in further analyses.

The prevalence of being simultaneously exposed to different combinations of challenging physical and psychosocial work exposures ranged between 7.1% and 13.7% among nurses and was somewhat higher among care assistants (11.0%–13.9%) (Table 3 and Table 4).

The results are divided into two different steps, according to the combinations. The first step concerns the effect between sickness absence and combinations of psychosocial job demands and heavy physical work and strenuous work postures. The second step describes sickness absence risks for combinations of job control and the two aspects of physical working conditions. Tables containing Model 1 and adjusted HRs for sickness absence complement descriptive figures of excess risks of the different combinations of physical and psychosocial exposures for sickness absence (Model 2). The results of the estimated additive interaction effects as measured in three different ways are presented in the same tables.

### 3.1. High Demands and Heavy Physical Work/Strenuous Postures

The excess risks for sickness absence among the individuals who reported working in a combination of high psychosocial demands and heavy physical work were higher among care assistants but also high among nurses (Figure 1a). Similar figures were found for those whose work included a combination of high psychosocial demands and strenuous postures (Figure 1b).

As shown in the detailed table (Table 3), the excess risk for sickness absence among those simultaneously being exposed to high demands and to heavy physical work was significant for care assistants (HR 1.51; CI 1.36–1.67), as was also the combination of high demands and strenuous postures (HR = 1.49; CI 1.36–1.63) (Table 3). This was the case for nurses also regarding the combination of high demands and heavy physical work (HR 1.43; CI 1.09–1.88) and regarding the combination of high demands and strenuous postures (HR = 1.42; CI 1.16–1.74).

The calculated synergy index for additive interaction effects was well above 1 in the combination of high job demands and heavy work in both occupational groups, but due to large confidence intervals, the synergy index was not significant (Table 3, upper part). The relative excess risk due to interaction (RERI), as well as the measurement of the attributable proportions (AP), were also insignificant in this case. This means that, although these two combinations, including high demands and heavy physical work and high demands and strenuous postures, mean an elevated risk for sickness absence, this excess risk was not higher than the sum of each of them.

### 3.2. Low Control and Heavy Physical Work/Strenuous Postures

The combination of low job control and heavy physical work did not significantly increase the risk of sickness absence among nurses, whereas the effect is strong among care assistants (Figure 2a). Figure 2b reveals that, among care assistants, low job control in itself (without having to work in strenuous postures) has a lowering effect on sickness absence, albeit very small (Figure 2b).

The combination of low job control and strenuous postures, however, was significant and relatively high both among nurses and care assistants: HR 1.45 CI; 1.17–1.80 and HR 1.42 CI; 1.30–1.56, respectively (Table 4). The synergy index (S) for the combination of low job control and heavy work was close to being significant among care assistants (S = 1.83; CI 0.96–3.47) (Table 4, upper part). Additionally, the other two indicators of synergetic effects (RERI and AP) were significant among care assistants, which indicates that avoiding situations of simultaneous exposure to low job control and heavy physical work would lower the sickness absence rate among these employees.

The synergy indexes (S) for the combination of low job control and strenuous postures were close to being significant among care assistants (S = 1.82 CI; 0.97–3.42) but not significant and low among nurses (S = 1.16 CI; 0.42–3.21) (Table 4, lower part). The other two indicators of additive synergetic effects (RERI and AP) were both significant among care assistants, which indicated that sickness absence might be reduced if the combination of lack of control and heavy work and the combination of lack of control and strenuous postures can be avoided (Table 4).

## 4. Discussion

Concerning the first aim of this study, the results showed that being simultaneously exposed to high psychosocial job demands and heavy physical work or high job demands and strenuous postures significantly increased the HRs for sickness absence among both nurses and care assistants when compared to employees not exposed to any of the factors in these combinations. Results with a similar magnitude of excess risks of sickness absence were found for the combinations of job control and heavy physical work and strenuous postures, respectively. These results are in-line with research reviews and previous studies on how combinations of physical and psychosocial exposures among health and social care workers affect health, sickness absence, or disability [24,26,27,29,30,31,32,33,49].

The second aim was to assess if the effect of the combinations was stronger than the sum of each of the components. Thus, we tested if the calculated measures of potential additive interactions reached significant levels. These estimations showed that the additive interaction effects regarding the combination of high job demands and heavy work were not significant in either occupational group. However, the additive interaction effects between the combination of job control and heavy physical work and the combination of job control and strenuous postures were significant among care assistants but not among nurses. This means that, although these two combinations, including high demands and heavy physical work and high demands and strenuous postures, constituted an elevated risk for sickness absence, this risk was not higher than the sum of each of them.

The main difference between nurses and care assistants was that two measures of relative excess risk due to interaction and attributable proportion (RERI and AP) were significant, and the synergy index (S) was close to significance among care assistants, but none of these measures were significant among nurses.

These partly contradictory results on synergetic effects among health and care employees reflect inconsistencies in previous research on these occupations, as well as on other occupations. Additive interaction effects of physical and psychosocial job factors on absenteeism or disability were reported in some studies [10,11,12,13,20,28], while a number of other studies found no additive interaction effects of such combined exposures on sickness absence [22,23,25,37].

The differences between nurses and care assistants observed in the present study may have methodological, as well as substantial, reasons. The fact that there were fewer nurses than care assistants in the study affected the statistical measurements, and the large confidence intervals also indicated heterogeneity among the employees of both groups. As shown in recent research, the working conditions among nurses, as well as among care assistants, are diverse with respect to work obligations and work organizations [24,25,26,27,28,30,31,32,33,34,36]. Some nurses and care assistants who reported both harmful psychosocial and physical challenges may be able to handle their activities in a planned way, while other employees have to work in urgent situations more regularly. Further, nurses are probably, to a higher degree than care assistants, able to foresee and avoid some situations that include heavy lifting or strenuous postures, even when their work is stressful. It is also possible that care assistants spend more time than nurses in combinations of stress and heavy physical work.

The fact that the signs of synergetic effects were weak in the present study may also be related to the question if the degree of demands or control can make much difference in the handling of heavy physical work in the daily work situation of female health and care workers. One study showed an increased risk for long-term sickness absence both among women with high job control and women without high control [7]. Similar implications were shown in two recent studies on burnout, one of which studied nurses, concluding that there were no buffering effects of the control on the negative effects of high psychosocial demands [50,51]. These studies point at a need for more detailed studies of how perceived high control among health and care employees may also increase the pressure on the employee to act in urgent situations where he or she does not have full information or full competence. As pointed out in recent systematic reviews, organizational factors such as downsizing and understaffing may increase the health risks of health and care employees [29,30].

For future policy-making and practices in health and social care, this study suggests that more attention should be given to how combinations of unfavorable physical and unfavorable psychosocial working conditions reinforce each other and increase the risks of sickness absence. The results particularly highlight the need for preventive action that aims to increase employee control over the workplace and participation in decision-making when heavy lifting and strenuous postures cannot be completely avoided through lifting aids, other technical measures, and cooperation instead of avoiding working alone. The Swedish Work Environment Act contains regulations related to individual work risks such as heavy work, work postures, psychosocial risks, and working alone but should be updated to include the excess risks of combinations of physical and psychosocial exposures.

### Strengths and Limitations

The two main advantages of the present study were its prospective design and the large population-based sample. The used survey questions in the SWES, as well as the measurement of sickness absence, have reasonable measurement qualities. Validity and reliability of the survey questions and the response alternatives were tested, and the survey was used in several previous studies. There are, however, also limitations related to heterogeneity of the large occupational groups and sex. The present study included nurses and care assistants, but a number of other health and care occupations were not studied: such as physicians, physiotherapists, psychologists, or administrators. Further, both nurses and care assistants work in a range of different organizational and physical and psychosocial contexts, which we unfortunately did not have information on. This includes working hours, working at a hospital versus working at people’s home, teamwork, shift work, degree of independence, and degree of specialization. Most of these and similar factors may influence how working conditions are reported by employees and, also, their associations to sickness absence. The fact that the gender distribution both among nurses and among care assistants was very skewed limited the possibilities to compare female and male employees and restricted conclusions concerning males. Our attempts to control for potential confounders were probably not enough to fully adjust for such differences.

Further, the reference category used in the present study (those who reported low exposure to physical and psychosocial risk factors) may be deviant in terms of its composition of individual factors that were not captured in the confounder control. It may, for example, contain larger proportions of employees who work part-time or who have supervisory positions, which we did not have information on.

The fact that the work environment exposures were measured only at one point in time may be an additional problem, since work environments change, and employees move between different workplaces and change their obligations. This is particularly important when the follow-up time is as long as three years after the year of inclusion, as it was in this study.

## 5. Conclusions

The practical implication of the present study is that large proportions of nurses and care assistants in Sweden were exposed to combinations of disadvantageous physical and psychosocial working conditions. These combinations increased the risk for future sickness absence in both occupational groups, and additive interaction effects of the combinations of low job control and heavy physical work and strenuous postures were found particularly among care assistants. In order to reduce these risks, lifting aids and measures to avoid working in bent or twisted postures are recommended, but also, improved psychosocial conditions are important, particularly better job control in terms of reduced time pressure and increased employee influence over work arrangements. Improved management and organizational measures, such as more staff in critical areas, may also reduce the risk for sickness absence.

The scientific lessons of the study are related to the shortage of previous studies designed to capture the fact that health and care workers often are exposed to simultaneous risks at work. Prospective studies looking at different combinations and different health outcomes are warranted, as well as studies focusing on theory development.

## Figures and Tables

**Figure 1 ijerph-17-07427-f001:**
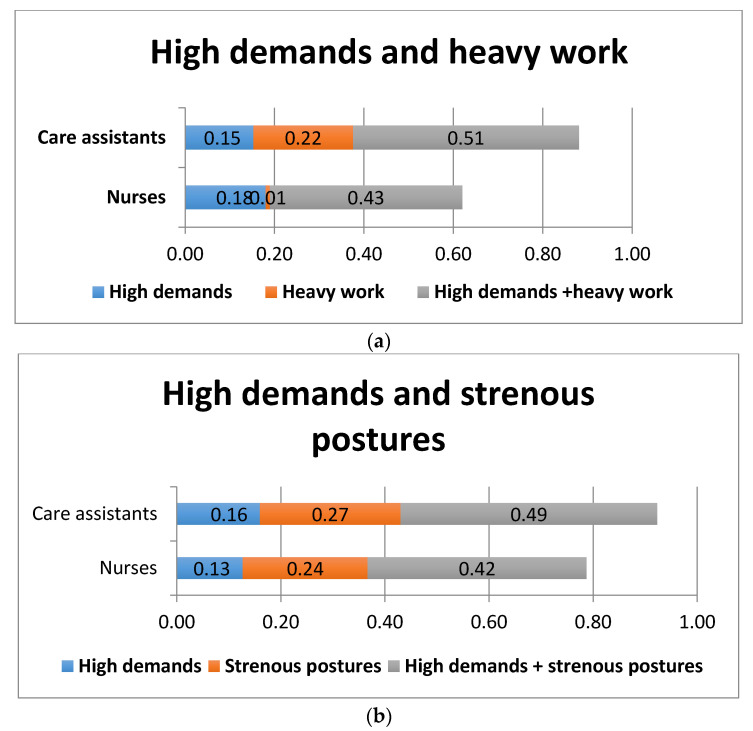
Excess risks for sickness absence among care assistants and nurses exposed to combinations of high psychosocial job demands and heavy physical work (**a**) or high psychosocial job demands and strenuous postures (**b**).

**Figure 2 ijerph-17-07427-f002:**
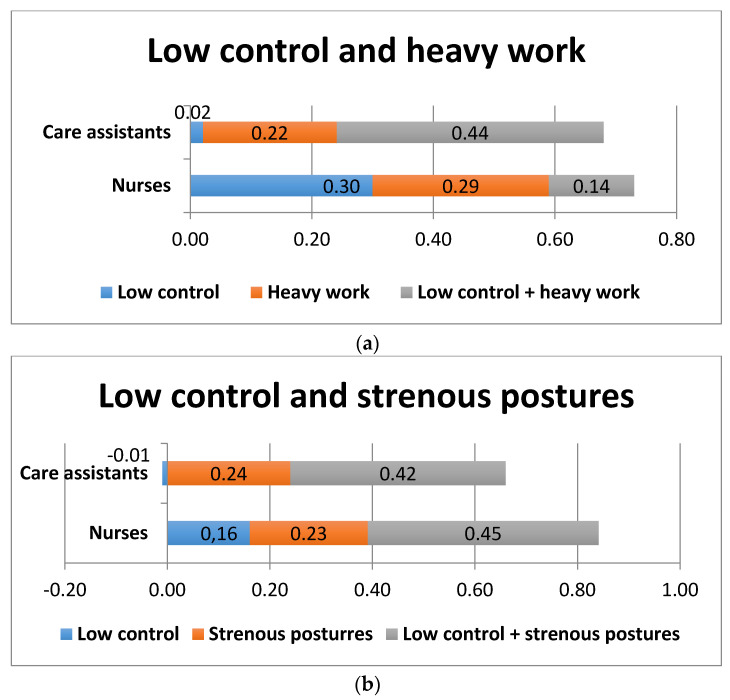
Excess risks for sickness absence among care assistants and nurses exposed to combinations of high psychosocial job control and heavy physical work (**a**) or high psychosocial job control and strenuous postures (**b**).

**Table 1 ijerph-17-07427-t001:** Number of participants in total and divided by compensated sickness absence (at least 15 days of total sickness absence in three years) or no sickness absence and by sex, according to age at interview and occupational group; 1994–2014 (*n* = 14,372).

	Total		No Sickness Absence				Sickness Absence			
	Men	Women	Men		Women		Men		Women	
	*n*	*n*	*n*	%	*N*	%	*n*	%	*n*	%
Total	1220	13,152	944	77.4	276	63.5	276	22.6	4795	36.5
Age (years)										
16–30	233	1742	209	89.7	1144	65.7	24	10.3	598	34.3
31–40	320	3235	262	81.9	2149	66.4	58	18.1	1086	33.6
41–50	336	4054	250	74.4	2610	64.4	86	25.6	1444	35.6
51–64	331	4121	223	67.4	2454	59.5	108	32.6	1667	40.5
Occupation										
Nursing professional	214	2502	166	77.6	1694	67.7	48	22.4	808	32.3
Care assistant	1006	10,650	778	77.3	6663	62.6	228	22.7	3987	37.4

**Table 2 ijerph-17-07427-t002:** Data on the selected confounders (sex, age, education, country of birth, and sector of emplyment); heavy work; strenuous postures; job demands; and job control among nurses and care assistants ^a^ stratified by no sickness absence and sickness absence (at least 15 days in three years).

	No Sickness Absence				Sickness Absence			
	Nurses		Care Assistants		Nurses		Care Assistants	
	*n* ^b^	*P* ^c^	*n* ^b^	*P* ^c^	*n* ^b^	*P* ^c^	*n* ^b^	*P* ^c^
Sex								
Men	166	8.9	778	10.5	48	5.6	228	5.4
Women	1694	91.1	6663	89.5	808	94.4	3987	94.6
Age								
16–29	138	7.4	1215	16.3	62	7.2	560	13.3
30–39	455	24.5	1956	26.3	180	21.0	964	22.9
40–49	652	35.1	2208	29.7	261	30.5	1269	30.1
50–64	615	33.1	2062	27.7	353	41.2	1422	33.7
Education								
>12 years	1846	99.2	1338	18.0	849	99.2	595	14.1
10–12 years	13	0.7	5511	74.1	6	0.7	3186	75.6
≤9 years	1	0.1	592	8.0	1	0.1	434	10.3
Country of birth								
Sweden	1745	93.8	6851	92.1	785	91.7	3799	90.2
Other country	115	6.2	589	7.9	71	8.3	415	9.8
Sector of employment								
Private organization	255	13.7	1330	17.9	108	12.7	656	15.6
Public organization	1602	86.3	6089	82.1	745	87.3	3552	84.4
Heavy work								
No	1267	86.5	4081	71.7	568	85.3	2048	63.5
Yes	197	13.5	1608	28.3	98	14.7	1175	36.5
Strenuous postures								
No	1277	78.0	4289	66.1	520	71.1	1994	55.0
Yes	361	22.0	2200	33.9	211	28.9	1631	45.0
Job demands								
Low	1100	60.4	5523	76.6	433	52.2	2882	70.4
High	721	39.6	1688	23.4	397	47.8	1213	29.6
Job control								
High	1263	69.7	5309	73.2	529	63.1	2825	68.7
Low	548	30.3	1945	26.8	309	36.9	1289	31.3

^a^ All incident cases of sickness absence diagnoses, including unspecified sickness absence (*n* = 14,372). ^b^ Number of individuals (*n*). ^c^ Prevalence (*P*) (%).

**Table 3 ijerph-17-07427-t003:** Co-exposure to job demands and heavy work or job demands and strenuous postures among nurses (*n* = 2716) and care assistants (*n* = 11,656) related to the hazard ratio (HR) and 95% confidence interval (CI) for long-term sickness absence ^a^.

		Nurses						Care Assistants				
	*n* ^b^	*P* ^c^	HR ^d^	HR ^e^	95% CI		*n* ^b^	*P* ^c^	HR ^d^	HR ^e^	95% CI	
**Job demands and heavy work ^f^**												
Low demands + not heavy work	294	50.4	1	1			1479	52.6	1	1		
High demands + not heavy work	258	35.8	**1.19**	**1.18**	1.02	1.38	526	16.1	**1.14**	**1.15**	1.05	1.27
Low demands + heavy work	35	6.2	0.97	0.99	0.70	1.40	703	20.4	**1.26**	**1.22**	1.12	1.33
High demands + heavy work	63	7.6	**1.46**	**1.43**	1.09	1.88	453	11.0	**1.55**	**1.51**	1.36	1.67
**Additive interaction**												
RERI ^g^				0.26	−0.26	0.77				0.13	0.06	0.33
AP ^h^				0.18	−0.15	0.51				0.09	0.04	0.21
S ^i^				2.45	0.23	25.80				1.35	0.84	2.16
**Job demands and strenuous postures ^f^**												
Low demands + not strenuous postures	289	46.9	1	1			1496	49.4	1	1		
High demands + not strenuous postures	220	29.1	1.14	1.13	0.96	1.33	455	12.7	**1.14**	**1.16**	1.05	1.29
Low demands + strenuous postures	82	10.3	1.25	1.24	0.98	1.58	986	24.6	**1.31**	**1.27**	1.18	1.37
High demands + strenuous postures	123	13.7	**1.44**	**1.42**	1.16	1.74	614	13.4	**1.54**	**1.49**	1.36	1.63
**Additive interaction**												
RERI ^g^				0.06	−0.35	0.46				0.06	−0.12	0.24
AP ^h^				0.04	−0.24	0.32				0.04	−0.08	0.16
S ^i^				1.15	0.40	3.34				1.13	0.76	1.70

^a^ All incident cases of sickness absence, including unspecified sickness absence (*n* = 14,372). ^b^ Number of cases (*n*). ^c^ Prevalence (*P*) of the exposure categories (%). ^d^ MODEL 1: hazard ratio (HR) and 95% confidence interval (CI), adjusted for age at interview (one-year intervals) and year of interview, significant figures are shown in bold (*p* < 0.05). ^e^ MODEL 2: Hazard ratio (HR) adjusted for age at interview (one-year intervals), year of interview, sex, education, sector of employment, and country of birth, significant figures are shown in bold (*p* < 0.05). ^f^ Four categories classifying the co-exposure to job demands and heavy work/strenuous postures. ^g^ Relative excess risk due to interaction (RERI). ^h^ Attributable proportions (AP), and ^i^ Rothman’s synergy index (S) and 95% confidence interval (CI), adjusted for age at interview (one-year intervals), year of interview, sex, education, sector of employment, and country of birth.

**Table 4 ijerph-17-07427-t004:** Co-exposure to job control and heavy physical work or job control and strenuous postures among nurses (*n* = 2716) and care assistants (*n* = 11,656) related to the hazard ratio (HR) and 95% confidence interval (CI) for long-term sickness absence ^a^.

		Nurses						Care Assistants				
	*n* ^b^	*P* ^c^	HR ^d^	HR ^e^	95% CI		*n* ^b^	*P* ^c^	HR ^d^	HR ^e^	95% CI	
**Job control and heavy work ^f^**												
High control + not heavy work	358	60.1	1	1			1460	50.6	1	1		
Low control + not heavy work	199	26.0	**1.31**	**1.30**	1.10	1.53	555	18.1	1.04	1.02	0.93	1.12
High control + heavy work	50	6.8	1.30	1.29	0.96	1.73	716	20.3	**1.26**	**1.22**	1.12	1.33
Low control + heavy work	46	7.1	1.16	1.14	0.83	1.55	441	11.0	**1.49**	**1.44**	1.30	1.60
**Additive interaction**												
RERI ^g^				−0.45	−0.98	0.09				**0.20**	0.01	0.38
AP ^h^				−0.39	−0.94	0.15				**0.14**	0.02	0.26
S ^i^				0.24	0.02	3.02				1.83	0.96	3.47
**Job control and strenuous postures ^f^**												
High control + not strenuous postures	351	54.6	1	1			1488	47.5	1	1		
Low control + not strenuous postures	162	21.3	1.18	1.16	0.97	1.39	482	14.7	1.00	0.99	0.90	1.09
High control + strenuous postures	103	12.9	**1.25**	1.23	0.99	1.53	968	24.0	**1.29**	**1.24**	1.15	1.34
Low control + strenuous postures	102	11.3	**1.47**	**1.45**	1.17	1.80	626	13.9	**1.47**	**1.42**	1.30	1.56
**Additive interaction**												
RERI ^g^				0.06	−0.36	0.48				**0.19**	0.03	0.36
AP ^h^				0.04	−0.24	0.33				**0.13**	0.02	0.25
S ^i^				1.16	0.42	3.21				1.82	0.97	3.42

^a^ All incident cases of sickness absence, including unspecified sickness absence (*n* = 14,372). ^b^ Number of cases (*n*). ^c^ Prevalence (*P*) of the exposure categories (%). ^d^ MODEL 1: hazard ratio (HR) and 95% confidence interval (CI), adjusted for age at interview (one-year intervals) and year of interview, significant figures are shown in bold (*p* < 0.05). ^e^ MODEL 2: hazard ratio (HR), adjusted for age at interview (one-year intervals), year of interview, sex, education, sector of employment, and country of birth, significant figures are shown in bold (*p* < 0.05). ^f^ Four categories classifying the co-exposure to job control and heavy work/strenuous postures. ^g^ Relative excess risk due to interaction (RERI). ^h^ Attributable proportions (AP), and ^i^ Rothman’s synergy index (S) and 95% confidence interval (CI), adjusted for age at interview (one-year intervals), year of interview, sex, education, sector of employment, and country of birth.

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
