# Peer review of "Interaction Effects of Physical and Psychosocial Working Conditions on Risk for Sickness Absence: A Prospective Study of Nurses and Care Assistants in Sweden"

_ijerph, 2020, doi:10.3390/ijerph17207427_

Round 1
Reviewer 1 Report
- We suggest that the methodology be clarified in the summary.
- Adequate introduction, with a clear demonstration of the relevance of the study based on the most recent evidence.
- Methodology clearly exposed and from what it is understood, this is a project of continuity of this research group which, from our perspective, values the work presented.
- The discussion is adequate to the objectives and data presented, however we would like to see a greater confrontation of authors.
- Proper conclusion.
Author Response
Dear Reviewer 1. Thank you very much for the valuable comments and suggestions, which have been very helpful in our revision of the manuscript. Please find our responses below.
Comments and Suggestions for Authors
- We suggest that the methodology be clarified in the summary.
Response: We have added a short sentence on methodology in the abstract
- Adequate introduction, with a clear demonstration of the relevance of the study based on the most recent evidence.
Response: Thank you!
- Methodology clearly exposed and from what it is understood, this is a project of continuity of this research group that, from our perspective, values the work presented.
Response: Thank you!
- The discussion is adequate to the objectives and data presented, however we would like to see a greater confrontation of authors.
Response: Thank you for pointing this out. We have now revised the discussion including some additional references to studies by other authors on interaction effects.
- Proper conclusion.
Response: Thank you!
Reviewer 2 Report
I appreciate the opportunity to review the "Interaction effects of physical and psychosocial working conditions on risk for sickness absence: A prospective study of nurses and care assistants in Sweden".
This study is a very interesting subject and a valuable research in terms of managing a hospital human resources management.
I think this study is well written. I just want authors to emphasize more in the introduction why this study is worth it.
Thank you for the authors' efforts.
Author Response
Dear Reviewer 2. Thank you very much for the valuable comments and suggestions, which have been very helpful in our revision of the manuscript. Please find our responses below.
Comments and Suggestions for Authors
I appreciate the opportunity to review the "Interaction effects of physical and psychosocial working conditions on risk for sickness absence: A prospective study of nurses and care assistants in Sweden".
-This study is a very interesting subject and a valuable research in terms of managing a hospital human resources management.
Response: Thank you!
-I think this study is well written. I just want authors to emphasize more in the introduction why this study is worth it.
Response: Thank you for pointing this out. We have now revised the introduction and added more information about why this study is of interest
-Thank you for the authors' efforts.
Reviewer 3 Report
REVIEWER COMMENTS:
Interaction effects of physical and psychosocial working conditions on
risk for sickness absence: A prospective study of nurses and care assistants
in Sweden
Thank you for the opportunity to review this manuscript. Overall, this is an interesting topic where the objective examines interaction effects of physical and psychosocial work conditions on future risk for sickness absence among nurses and care assistants in Sweden.
I don't feel qualified to judge about the English language and style. I sense that there is good data in this work, however, the presentation and interpretation of the study need significant additional thought and up-to-date references. Here are some suggestions to improve the manuscript:
First, In the Abstract section, please consider introduce some additional information about the significance of the results (for example: the Test and p).
- Introduction
Why does the introduction start talking about slaughterhouse workers? If there is too much literature that deals with the working conditions of health professionals?
The introduction doesn´t provide sufficient background around physical and psychosocial work conditions in clinical places. You can include more relevant and new references (last years).
It would have been helpful to provide some degree of conceptual framework to help the reader understand the consideration of the .
Please, explain if your variables have been used in others studies and the review literature.
The effects of physical and psychosocial work conditions on risk for sickness absence have been previously and widely studied, What does your study add new compared to others already known?
- Materials and Methods
How was the data collected? Is it a validated questionnaire? Was the ratio of men to women comparable?
- Results
The differences between the two occupational groups 216 regarding sickness absence were smaller among men than among women (Table 1). You should indicate with which test you have tested it and the significance.
It is very difficult to understand the tables of the results.
The presentation of Table 2 would improve the interpretation of the results.
I do not believe that Figure 1 and Figure 2 provides new information regarding Tables.
- Discussion
The discussion needs greater clarity on what the results showed.
- Start the discussion by reporting your own findings from the present study and then, after that, you put it in perspective of other available research and please, write new references. The discussion fails to clearly attempt to identify/explain reasons for results in this study that differ from other studies looking at similar outcomes. You have only 9 references with less than 5 years, of the 40 that you present in all the manuscript.
Please include more Limitations of this study. For example, the number of men and women was very uneven, which could have influenced the results; the wide range of professions that healthcare providers can include. You do not specify if the questionnaire is validated…..
Please include a interesting Practical implications section before Conclusion section
- Conclusion should state only your findings according your objectives. Reconsider the conclusion to align more clearly with the aim and results of the study.
Not all sentences are conclusions of your research. For example: “In order to reduce these risks, lifting aids and measures to avoid work in bent or twisted postures are recommended, but also improved psychosocial conditions are important, and particularly better job control in terms of reduced time pressure and increased employee influence over work arrangements”.
Author Response
Dear Reviewer 3. Thank you very much for the valuable comments and suggestions, which have been very helpful in our revision of the manuscript. Please find our responses below.
REVIEWER COMMENTS:
Interaction effects of physical and psychosocial working conditions on
risk for sickness absence: A prospective study of nurses and care assistants
in Sweden
Thank you for the opportunity to review this manuscript. Overall, this is an interesting topic where the objective examines interaction effects of physical and psychosocial work conditions on future risk for sickness absence among nurses and care assistants in Sweden.
I don't feel qualified to judge about the English language and style. I sense that there is good data in this work, however, the presentation and interpretation of the study need significant additional thought and up-to-date references. Here are some suggestions to improve the manuscript:
First, In the Abstract section, please consider introduce some additional information about the significance of the results (for example: the Test and p).
Response: Good suggestion, we have now included more information about HR and CI in the abstract.
Introduction
Why does the introduction start talking about slaughterhouse workers? If there is too much literature that deals with the working conditions of health professionals?
Response: We have now restructured the introduction and started with a section on sickness absence and working conditions and also made the section on health professionals more separate from studies on other occupations. As the study of slaughterhouse workers was one of the earliest to point at interaction effects between physical and psychosocial factors we have kept the reference.
The introduction doesn´t provide sufficient background around physical and psychosocial work conditions in clinical places. You can include more relevant and new references (last years).
Response: We have now updated the introduction and added relevant research reviews and studies on physical and psychosocial factors affecting health and care workers health and sickness absence
It would have been helpful to provide some degree of conceptual framework to help the reader understand the consideration of the .
Response: We have not been able to find any other theoretical or conceptual framework that is dealing with interaction effects than those established around psychosocial factors eg the demand/support model and the effort/reward imbalance model. Maybe this reflects that epidemiology in general is poor in theoretical and conceptual development.
Please, explain if your variables have been used in others studies and the review literature.
Response: Our research team has used the SWES data and its variables in several studies. The most recent and relevant for the present study have been mentioned in the manuscript (Gustafsson, K. et al 2020, Marklund, S. et al 2019, Leineweber, C. 2019). We are not aware that any of these studies have been included in a review.
The effects of physical and psychosocial work conditions on risk for sickness absence have been previously and widely studied. What does your study add new compared to others already known?
Response: As we argue in the introduction, most studies on physical and psychosocial work factors affecting risk for sickness absence focus on single factor rather than on interaction between factors. This is true for studies of different occupations but is particularly problematic among health and care employees as it is well known that many employees are simultaneously exposed to both physical and psychosocial risk factors. We have found no studies at all which have investigated such interaction with regard to sickness absence.
- Materials and Methods
How was the data collected? Is it a validated questionnaire? Was the ratio of men to women comparable?
Response: Thank you for pointing out this important issue. We have now added more information in the method section about the data collection and about the validation of the SWES questionnaire.
In the study group 8.5% were men 91.5% were women. The distribution in the sample is similar to that of the employed population, but as there are so few men we have pointed out in the limitations section that we cannot draw any conclusions for males in this study. However, we have adjusted for sex in model 2 in the tables.
- Results
The differences between the two occupational groups regarding sickness absence were smaller among men than among women (Table 1). You should indicate with which test you have tested it and the significance.
Response: In table 1, we only report the study group descriptively – which means that no significant test is needed here. In continued analyzes, we adjust for sex (confounder) (see method and footnote table 3 and 4, model 2)
It is very difficult to understand the tables of the results.
Response: We have now clarified the tables (see table 3 and 4).
The presentation of Table 2 would improve the interpretation of the results.
Response: We have clarified this (see table 2).
I do not believe that Figure 1 and Figure 2 provides new information regarding Tables.
Response: We agree with the reviewer, but as we believe that the figures are easier interpret than the tables, and as none of the other four reviewers have suggested any changes in this respect we have kept the figures for pedagogical reasons.
- Discussion
The discussion needs greater clarity on what the results showed.
- Start the discussion by reporting your own findings from the present study and then, after that, you put it in perspective of other available research and please, write new references. The discussion fails to clearly attempt to identify/explain reasons for results in this study that differ from other studies looking at similar outcomes. You have only 9 references with less than 5 years, of the 40 that you present in all the manuscript.
Response: Yes, we agree and have followed your suggestion concerning structure and added a number of more recent references.
Please include more Limitations of this study. For example, the number of men and women was very uneven, which could have influenced the results; the wide range of professions that healthcare providers can include. You do not specify if the questionnaire is validated.
Response: In the study group 8.5% were men 91.5% were women. The distribution in the sample is similar to that of the employed population, but as there are so few men we have pointed out in the limitations section that we cannot draw any conclusions for males in this study. However, we have adjusted for sex in model 2. The point on heterogeneity of health care professions has been added to the limitations section. We have now clarified the point on validation in the methods section.
Please include a interesting Practical implications section before Conclusion section
Response: Thank you for pointing out the difference between practical and scientific conclusions. We have now done as you suggested.
- Conclusion should state only your findings according your objectives. Reconsider the conclusion to align more clearly with the aim and results of the study.
Response: We agree and have been stricter here.
Not all sentences are conclusions of your research. For example: “In order to reduce these risks, lifting aids and measures to avoid work in bent or twisted postures are recommended, but also improved psychosocial conditions are important, and particularly better job control in terms of reduced time pressure and increased employee influence over work arrangements”.
Response: As suggested in an earlier comment we have now more carefully separated practical and scientific implications.
Reviewer 4 Report
Dear authors,
Thank you for your interesting submission! Please find my comments below to improve your manuscript.
1. Please try your best to find out one theory to explain the relationship between physical and psychological factors and absent behaviour, and further literature review regarding this theory among health professions. This is very important to the contributions of this study.
2. Please clarify the sample you wanted to focus on. For me, health professions in some countries all over the world reported lower absent rates than the populations in this study. You have to focus on one particular sample due to specific reasons. Thus, further revision or editing on your research question is also necessary.
3. Have you ever closed any research gap based on your literature review on the previous studies and relative studies of the theory you wanted to apply in your own study? Clarify your contributions to this research field.
4. How could you ensure the quality of your sampling? I believe more information on the Swedish Work Environment Survey is necessary.
Have you ever made any additional action for your sampling and further measures in order to enhance the quality of your collected information?
5. How did you ensure the quality of your instruments in this study?
6. How did you control the confounders and bias in your statistical analysis?
Good luck!
Author Response
Dear Reviewer 4. Thank you very much for the valuable comments and suggestions, which have been very helpful in our revision of the manuscript. Please find our responses below.
Comments and Suggestions for Authors
Dear authors,
Thank you for your interesting submission! Please find my comments below to improve your manuscript.
- Please try your best to find out one theory to explain the relationship between physical and psychological factors and absent behaviour, and further literature review regarding this theory among health professions. This is very important to the contributions of this study.
Response: We have not been able to find any other theoretical or conceptual framework that is dealing with interaction effects than those established around psychosocial factors eg the demand/support model and the effort/reward imbalance model. Maybe this reflects that epidemiology in general is poor in theoretical and conceptual development.
- Please clarify the sample you wanted to focus on. For me, health professions in some countries all over the world reported lower absent rates than the populations in this study. You have to focus on one particular sample due to specific reasons. Thus, further revision or editing on your research question is also necessary.
Response: In Sweden, health professionals have historically had lower sickness absence rates that the average working population, but since the late 1990s it has been higher. We have clarified this in the introduction and in the aims and results of the study.
- Have you ever closed any research gap based on your literature review on the previous studies and relative studies of the theory you wanted to apply in your own study? Clarify your contributions to this research field.
Response: We have restructured the introduction and discussion in order to reach more clarity on the background as well as on the contribution of our study to the research field
- How could you ensure the quality of your sampling? I believe more information on the Swedish Work Environment Survey is necessary.
Have you ever made any additional action for your sampling and further measures in order to enhance the quality of your collected information?
Response: We have now included more information in the methods section about the issues sampling and quality of the information
- How did you ensure the quality of your instruments in this study?
Response: We have now added more information about this in the methods section.
- How did you control the confounders and bias in your statistical analysis?
Response: Yes, additionally, the associations were analyzed by an extension of the adjustment to also include sex, education, country of birth, and sector of employment (Model 2 in table 3 and 3).
Good luck!
Reviewer 5 Report
A well-structured and well-documented draft has yielded unsurprising results in terms of the risk of combined physical and psychosocial strain on health and social workers. The conclusions are also in place, although they could have gone further and proposed management measures, e.g. increase the number of employees in critical places.
Author Response
Dear Reviewer 5. Thank you very much for the valuable comments and suggestions, which have been very helpful in our revision of the manuscript. Please find our responses below.
Comments and Suggestions for Authors
A well-structured and well-documented draft has yielded unsurprising results in terms of the risk of combined physical and psychosocial strain on health and social workers. The conclusions are also in place, although they could have gone further and proposed management measures, e.g. increase the number of employees in critical places.
Response: Thank you for this suggestion. We have added this aspect in the conclusion and also added a sentence on this issue in the discussion in relationship to the review by Bernal et al 2015.
Round 2
Reviewer 3 Report
Thank you very much to improve the manuscript. Congratulations!
Author Response
Thank you very much for the comments and for taking the time to review the manuscript.
Reviewer 4 Report
Dear authors,
Although "epidemiology in general is poor in theoretical and conceptual development", necessary information to construct your proposed model is needed. For instance, how the job demand-control model explains your model should be added in your manuscript (lines 55-57). Moreover, are you sure the proposed model could be explained by the effort-reward balance model?
Additionally, it is necessary to present which revision you have made in your response letter. It could help the reviewers understand where the revision is.
Finally, further implications for future policies-making and practices should also be added in your discussion section.
Author Response
Please see the attachment.

This manuscript is a resubmission of an earlier submission. The following is a list of the peer review reports and author responses from that submission.